# Fixed Point Theorems for Geraghty Contraction Type Mappings in *b*-Metric Spaces and Applications

**Hamid Faraji [1], Dragana Savić [2] and Stojan Radenović [3,4,*]**

[1] Department of Mathematics, College of Technical and Engineering, Saveh Branch, Islamic Azad University, Saveh 39187/366, Iran; faraji@iau-saveh.ac.ir
[2] Primari School "Kneginja Milica", Beograd 11073, Serbia; gagasavic89@gmail.com
[3] Nonlinear Analysis Research Group, Ton Duc Thang University, Ho Chi Minh City 700000, Vietnam
[4] Faculty of Mathematics and Statistics, Ton Duc Thang University, Ho Chi Minh City 700000, Vietnam
[*] Correspondence: stojan.radenovic@tdtu.edu.vn

**Abstract:** In this paper, some new results are given on fixed and common fixed points of Geraghty type contractive mappings defined in *b*-complete *b*-metric spaces. Moreover, two examples are represented to show the compatibility of our results. Some applications for nonlinear integral equations are also given.

**Keywords:** fixed point; Geraghty; *b*-metric space

## 1. Introduction

In 1989, Bakhtin [1] introduced b-metric spaces as a generalization of metric spaces. Since then, several papers have been published on the fixed point theory in such spaces. For further works and results in *b*-metric spaces, we refer readers to References [2–22].

**Definition 1.** *Let X be a (nonempty) set and $s \geq 1$ be a given real number. A function $d : X \times X \to [0, \infty)$ is called a b-metric on X if the following conditions hold for all $x, y, z \in X$:*

(i) $d(x, y) = 0$ *if and only if* $x = y$,
(ii) $d(x, y) = d(y, x)$,
(iii) $d(x, y) \leq s[d(x, z) + d(z, y)]$ *(b-triangular inequality).*

*Then, the pair $(X, d)$ is called a b-metric space with parameter s.*

**Example 1.** *[14] Let $(X, d)$ be a metric space and let $\beta > 1, \lambda \geq 0$ and $\mu > 0$. For $x, y \in X$, set $\rho(x, y) = \lambda d(x, y) + \mu d(x, y)^{\beta}$. Then $(X, \rho)$ is a b-metric space with the parameter $s = 2^{\beta - 1}$ and not a metric space on X.*

In 1973, Geraghty [23] introduced a class of functions to generalize the Banach contraction principle. Let $S$ be the family of all functions $\alpha : [0, \infty) \to [0, 1)$ satisfying the property:

$$\lim_{n \to \infty} \alpha(t_n) = 1 \quad implies \quad \lim_{n \to \infty} t_n = 0.$$

**Theorem 1.** *[23] Let $(X, d)$ be a complete metric space. Let $T : X \to X$ be given mapping satisfying:*

$$d(Tx, Ty) \leq \alpha(d(x, y))d(x, y), \quad x, y \in X,$$

*where $\alpha \in S$. Then T has a unique fixed point.*

In 2011, Dukic et al. [24] reconsidered Theorem 1 in the framework of $b$-metric spaces (see also Reference [25]).

Let $(X, d)$ be a $b$-metric space with parameter $s \geq 1$ and $S$ denote the set of all functions $\alpha : [0, \infty) \to [0, \frac{1}{s})$, satisfying the following condition:

$$\lim_{n \to \infty} \alpha(t_n) = \frac{1}{s} \Rightarrow \lim_{n \to \infty} t_n = 0.$$

**Theorem 2.** *[24] Let $(X, d)$ be a $b$-complete $b$-metric space with parameter $s \geq 1$ and let $T : X \to X$ be a self-map. Suppose that there exists $\beta \in S$ such that:*

$$d(Tx, Ty) \leq \beta(d(x, y))d(x, y),$$

*holds for all $x, y \in X$. Then $T$ has a unique fixed point $x^* \in X$.*

In recent years, many researchers have extended the result of Geraghty in the context of various metric spaces (e.g., see References [26–29]). In the present paper, we extended some fixed point theorems for Geraghty contractive mappings in $b$-metric spaces.

## 2. Results

Let $\mathcal{B}$ denote the set of all functions $\beta : [0, \infty) \to [0, \frac{1}{s})$ which satisfies the condition $\limsup_{n \to \infty} \beta(t_n) = \frac{1}{s}$ implies that $t_n \to 0$ as $n \to \infty$ [25].

**Theorem 3.** *Let $(X, d)$ be a $b$-complete $b$-metric space with parameter $s \geq 1$. Let $T : X \to X$ be a self-mapping satisfying:*

$$d(Tx, Ty) \leq \beta(M(x, y))M(x, y), \quad x, y \in X, \tag{1}$$

*where:*

$$M(x, y) = max\{d(x, y), d(x, Tx), d(y, Ty), \frac{1}{2s}(d(x, Ty) + d(y, Tx))\},$$

*and $\beta \in \mathcal{B}$. Then $T$ has a unique fixed point.*

**Proof of Theorem 3.** Let $x_0 \in X$ be arbitrary. Consider the sequence $\{x_n\}$ where:

$$x_n = Tx_{n-1} = T^n x_0, \quad n \in \mathbb{N}.$$

If there exists $n \in \mathbb{N}$ such that $x_{n+1} = x_n$, then $x_n$ is a fixed point of $T$ and the proof is finished. Otherwise, we have $d(x_{n+1}, x_n) > 0$ for all $n \in \mathbb{N}$. By Condition (1), for all $n \in \mathbb{N}$ we have:

$$d(x_n, x_{n+1}) = d(Tx_{n-1}, Tx_n) \leq \beta(M(x_{n-1}, x_n))M(x_{n-1}, x_n), \tag{2}$$

where:

$$
\begin{aligned}
M(x_{n-1}, x_n) &= max\{d(x_{n-1}, x_n), d(x_{n-1}, Tx_{n-1}), d(x_n, Tx_n), \frac{d(x_{n-1}, Tx_n) + d(x_n, Tx_{n-1})}{2s}\} \\
&= max\{d(x_{n-1}, x_n), d(x_{n-1}, x_n), d(x_n, x_{n+1}), \frac{d(x_{n-1}, x_{n+1}) + d(x_n, x_n)}{2s}\} \\
&\leq max\{d(x_{n-1}, x_n), d(x_n, x_{n+1}), \frac{s(d(x_{n-1}, x_n) + d(x_n, x_{n+1}))}{2s}\} \\
&= max\{d(x_{n-1}, x_n), d(x_n, x_{n+1})\}.
\end{aligned}
$$

If $d(x_{n-1}, x_n) \leq d(x_n, x_{n+1})$, then $M(x_{n-1}, x_n) = d(x_n, x_{n+1})$. From Condition (2), we have:

$$d(x_n, x_{n+1}) \leq \beta(M(x_{n-1}, x_n))M(x_{n-1}, x_n)$$
$$\leq \frac{1}{s}d(x_n, x_{n+1}) \quad n \in \mathbb{N}.$$

This is a contradiction. Thus, we have:

$$M(x_{n-1}, x_n) = d(x_n, x_{n-1})$$

.

Then, from Condition (2), we get:

$$d(x_n, x_{n+1}) \quad \leq \quad \beta(M(x_{n-1}, x_n))d(x_{n-1}, x_n) \tag{3}$$
$$< \quad d(x_{n-1}, x_n), \quad n \in \mathbb{N}.$$

So $\{d(x_{n-1}, x_n)\}$ is a decreasing sequence of non-negative reals. Hence, there exists $r \geq 0$ such that $d(x_{n-1}, x_n) \to r$ as $n \to \infty$. We claimed that $r = 0$. Suppose on the contrary that $r > 0$, then from Condition (3), we have:

$$r \leq \limsup_{n \to \infty} \beta(M(x_{n-1}, x_n))r.$$

Then,

$$\frac{1}{s} \leq 1 \leq \limsup_{n \to \infty} \beta(M(x_{n-1}, x_n)) \leq \frac{1}{s}.$$

Since $\beta \in \mathcal{B}$, then $\lim_{n \to \infty} M(x_{n-1}, x_n) = 0$. So $\lim_{n \to \infty} d(x_{n-1}, x_n) = 0$, which is a contradiction, that is, $r = 0$. Now we show that $\{x_n\}$ is a $b$-Cauchy sequence. Suppose on the contrary that $\{x_n\}$ is not a $b$-Cauchy sequence. Then there exists $\varepsilon > 0$ for which we can find subsequences $\{x_{m(k)}\}$ and $\{x_{n(k)}\}$ of $\{x_n\}$ such that $n(k)$ is the smallest index for which $n(k) > m(k) > k$,

$$d(x_{m(k)}, x_{n(k)}) \geq \varepsilon, \tag{4}$$

and

$$d(x_{m(k)}, x_{n(k)-1}) < \varepsilon. \tag{5}$$

From Condition (5) and using the $b$-triangular inequality, we have:

$$\varepsilon \leq d(x_{m(k)}, x_{n(k)}) \leq s(d(x_{m(k)}, x_{m(k)+1}) + d(x_{m(k)+1}, x_{n(k)})).$$

Then, we get:

$$\frac{\varepsilon}{s} \leq \limsup_{k \to \infty} d(x_{m(k)+1}, x_{n(k)}). \tag{6}$$

Therefore,

$$\limsup_{k\to\infty} M(x_{m(k)}, x_{n(k)-1}) = \limsup_{k\to\infty} max\{d(x_{m(k)}, x_{n(k)-1}), d(x_{m(k)}, Tx_{m(k)}),$$

$$d(x_{n(k)-1}, Tx_{n(k)-1}), \frac{d(x_{m(k)}, Tx_{n(k)-1}) + d(x_{n(k)-1}, Tx_{m(k)})}{2s}\}$$

$$= \limsup_{k\to\infty} max\{d(x_{m(k)}, x_{n(k)-1}), d(x_{m(k)}, x_{m(k)+1}),$$

$$d(x_{n(k)-1}, x_{n(k)}), \frac{d(x_{m(k)}, x_{n(k)}) + d(x_{n(k)-1}, x_{m(k)+1})}{2s}\}$$

$$\leq \limsup_{k\to\infty} max\{d(x_{m(k)}, x_{n(k)-1}), d(x_{m(k)}, x_{m(k)+1}), d(x_{n(k)-1}, x_{n(k)}),$$

$$\frac{sd(x_{m(k)}, x_{n(k)-1}) + sd(x_{n(k)}, x_{n(k)-1})}{2s}$$

$$+ \frac{sd(x_{n(k)-1}, x_{m(k)}) + sd(x_{m(k)}, x_{m(k)+1})}{2s}\}$$

$$\leq \varepsilon.$$

From Condition (6) and Condition (1), we have:

$$\frac{\varepsilon}{s} \leq \limsup d(x_{m(k)+1}, x_{n(k)})$$

$$\leq \limsup_{k\to\infty} \beta(M(x_{m(k)}, x_{n(k)-1}))M(x_{m(k)}, x_{n(k)-1})$$

$$\leq \varepsilon \limsup_{k\to\infty} \beta(M(x_{m(k)}, x_{n(k)-1})).$$

Then $\frac{1}{s} \leq \limsup_{k\to\infty} \beta(M(x_{m(k)}, x_{n(k)-1})) \leq \frac{1}{s}$. Since $\beta \in \mathcal{B}$, so $M(x_{m(k)}, x_{n(k)-1}) \to 0$, as a result, $d(x_{m(k)}, x_{n(k)-1}) \to 0$. From Condition (4) and using the $b$-triangular inequality, we have:

$$\varepsilon \leq d(x_{m(k)}, x_{n(k)}) \leq s(d(x_{m(k)}, x_{n(k)-1}) + d(x_{n(k)-1}, x_{n(k)})).$$

Therefore, $\lim_{k\to\infty} d(x_{m(k)}, x_{n(k)}) = 0$. This contradicts with Condition (4). Hence, $\{x_n\}$ is a $b$-Cauchy sequence. The completeness of $X$ implies that there exists $u \in X$ such that $x_n \to u$. We showed that $u$ is a fixed point of $T$. By $b$-triangular inequality and Condition (1), we have:

$$d(u, Tu) \leq s(d(u, Tx_n) + d(Tx_n, Tu))$$

$$\leq sd(u, Tx_n) + s\beta(M(x_n, u))M(x_n, u).$$

Letting $n \to \infty$ in the above inequality, we obtain:

$$d(u, Tu) \leq s\limsup_{n\to\infty} d(u, x_{n+1}) \tag{7}$$

$$+ s\limsup_{n\to\infty} \beta(M(x_n, u)) \limsup_{n\to\infty} M(x_n, u),$$

where:

$$\limsup_{n\to\infty} M(x_n, u) = \limsup_{n\to\infty} max\{d(x_n, u), d(x_n, Tx_n), d(u, Tu), \frac{1}{2s}(d(x_n, Tu) + d(u, Tx_n))\}$$

$$\leq \limsup_{n\to\infty} max\{d(x_n, u), d(x_n, x_{n+1}), d(u, Tu), \frac{1}{2s}(sd(x_n, u) + sd(u, Tu) + d(u, x_{n+1}))\}$$

$$\leq d(u, Tu).$$

Hence, from Condition (7), we have:

$$d(u, Tu) \leq s\limsup \beta(M(x_n, u))d(u, Tu).$$

Consequently, $\frac{1}{s} \leq \limsup_{n\to\infty} \beta(M(x_n, u)) \leq \frac{1}{s}$. Since $\beta \in \mathcal{B}$, we concluded $\lim_{n\to\infty} M(x_n, u) = 0$.

Therefore, $Tu = u$. To see that the fixed point $u \in X$ is unique, suppose there is $v \neq u$ in $X$ such that $Tv = v$. From Condition (1), we get:

$$d(u, v) = d(Tu, Tv) \leq \beta(M(u, v))M(u, v),$$

where:

$$
\begin{aligned}
M(u, v) &= max\{d(u, v), d(u, Tu), d(v, Tv), \frac{1}{2s}(d(u, Tv) + d(v, Tu))\} \\
&\leq d(u, v).
\end{aligned}
$$

Therefore, we have $d(u, v) < \frac{1}{s}d(u, v)$. Then $u = v$, which is a contradiction. $\square$

**Example 2.** *Let $X = \{1, 2, 3\}$ and $d : X \times X \to [0, \infty)$ be defined as follows:*

(i) $d(1, 2) = d(2, 1) = 1$,
(ii) $d(1, 3) = d(3, 1) = \frac{1}{9}$,
(iii) $d(2, 3) = d(3, 2) = \frac{6}{9}$.
(iv) $d(1, 1) = d(2, 2) = d(3, 3) = 0$.

*It is easy to check that $(X, d)$ is a b-metric space with constant $s = \frac{3}{2}$. Set $T1 = T3 = 1, T2 = 3$ and $\beta(t) = \frac{2}{3}e^{-t}, t > 0$ and $\beta(0) \in [0, \frac{2}{3})$. Then we have:*

$$d(T1, T2) = d(1, 3) = \frac{1}{9} \leq \frac{2}{3}e^{-1} = \beta(M(1, 2))M(1, 2),$$

$$d(T1, T3) = d(1, 1) = 0 \leq \beta(M(1, 3))M(1, 3),$$

$$d(T2, T3) = d(3, 1) = \frac{1}{9} \leq \frac{2}{3}e^{-\frac{6}{9}}\left(\frac{6}{9}\right) = \beta(M(2, 3))M(2, 3).$$

*Therefore, the conditions of Theorem 3 are satisfied.*

**Theorem 4.** *Let $(X, d)$ be a b-complete b-metric space with parameter $s \geq 1$. Let $T, S$ be self-mappings on $X$ which satisfy:*

$$sd(Tx, Sy) \leq \beta(M(x, y))M(x, y), \quad x, y \in X, \tag{8}$$

*where $M(x, y) = max\{d(x, y), d(x, Tx), d(y, Sy)\}$ and $\beta \in \mathcal{B}$. If $T$ or $S$ are continuous, then $T$ and $S$ have a unique common fixed point.*

**Proof of Theorem 4.** Let $x_0$ be arbitrary. Define the sequence $\{x_n\}$ in $X$ by $x_{2n+1} = Tx_{2n}$ and $x_{2n+2} = Sx_{2n+1}$ for all $n = 0, 1, \ldots$. From Condition (8), for all $n = 0, 1, 2, \ldots$, we have:

$$
\begin{aligned}
sd(x_{2n+1}, x_{2n+2}) &= sd(Tx_{2n}, Sx_{2n+1}) \\
&\leq \beta(M(x_{2n}, x_{2n+1}))M(x_{2n}, x_{2n+1}),
\end{aligned}
\tag{9}
$$

where:

$$
\begin{aligned}
M(x_{2n}, x_{2n+1}) &= max\{d(x_{2n}, x_{2n+1}), d(x_{2n}, Tx_{2n}), d(x_{2n+1}, Sx_{2n+1})\} \\
&= max\{d(x_{2n}, x_{2n+1}), d(x_{2n+1}, x_{2n+2})\}, \quad n = 0, 1, 2, \ldots.
\end{aligned}
$$

If $M(x_{2n}, x_{2n+1}) = d(x_{2n+1}, x_{2n+2})$, then:

$$sd(x_{2n+1}, x_{2n+2}) \leq \beta(M(x_{2n}, x_{2n+1}))d(x_{2n+1}, x_{2n+2}) < \frac{1}{s}d(x_{2n+1}, x_{2n+2}),$$

which is a contradiction. Hence, we have $M(x_{2n}, x_{2n+1}) = d(x_{2n}, x_{2n+1})$. From Condition (9), we have:

$$
\begin{aligned}
d(x_{2n+1}, x_{2n+2}) &\leq \beta(M(x_{2n}, x_{2n+1}))d(x_{2n}, x_{2n+1}) \\
&\leq \frac{1}{s}d(x_{2n}, x_{2n+1}).
\end{aligned}
\tag{10}
$$

Then, we get $d(x_{2n+1}, x_{2n+2}) \leq d(x_{2n}, x_{2n+1})$. Similarly, $d(x_{2n+3}, x_{2n+2}) \leq d(x_{2n+2}, x_{2n+1})$. So, we have $d(x_n, x_{n+1}) \leq d(x_{n-1}, x_n)$. Thus $\{d(x_n, x_{n+1})\}$ is a nonincreasing sequence, hence there exists $r \geq 0$ such that $d(x_n, x_{n+1}) \to r$ as $n \to \infty$. We showed that $r = 0$. Suppose on the contrary that $r > 0$. Letting $n \to \infty$ in (10), we obtain:

$$
r \leq \limsup_{n \to \infty} \beta(M(x_{2n}, x_{2n+1}))r.
$$

Then, we have:

$$
\frac{1}{s} \leq 1 \leq \limsup_{n \to \infty} \beta(M(x_{2n}, x_{2n+1})) \leq \frac{1}{s}.
$$

Since $\beta \in \mathcal{B}$, we have:

$$
\lim_{n \to \infty} M(x_{2n}, x_{2n+1}) = 0.
$$

Hence,

$$
r = \lim_{n \to \infty} d(x_{2n}, x_{2n+1}) = 0,
$$

which is a contradiction. Now, we show that $\{x_{2n}\}$ is a $b$-Cauchy sequence. Suppose that $\{x_{2n}\}$ is not a $b$-Cauchy sequence. Then there exists $\varepsilon > 0$ for which we can find subsequences $\{x_{2m(k)}\}$ and $\{x_{2n(k)}\}$ of $\{x_{2n}\}$ such that $n(k)$ is the smallest index for which $n(k) > m(k) > k$,

$$
d(x_{2n(k)}, x_{2m(k)}) \geq \varepsilon,
\tag{11}
$$

and

$$
d(x_{2n(k)}, x_{2m(k)-2}) < \varepsilon.
\tag{12}
$$

From Condition (8) and Condition (11) and the $b$-triangular inequality, we have:

$$
\begin{aligned}
\varepsilon &\leq d(x_{2n(k)}, x_{2m(k)}) \\
&\leq sd(x_{2n(k)}, x_{2n(k)+1}) + sd(x_{2n(k)+1}, x_{2m(k)}) \\
&= sd(x_{2n(k)}, x_{2n(k)+1}) + sd(Tx_{2n(k)}, Sx_{2m(k)-1}) \\
&\leq sd(x_{2n(k)}, x_{2n(k)+1}) \\
&\quad + \beta(M(x_{2n(k)}, x_{2m(k)-1}))M(x_{2n(k)}, x_{2m(k)-1}),
\end{aligned}
\tag{13}
$$

where:

$$
M(x_{2n(k)}, x_{2m(k)-1}) = max\{d(x_{2n(k)}, x_{2m(k)-1}), d(x_{2n(k)}, Tx_{2n(k)}), d(x_{2m(k)-1}, Sx_{2m(k)-1})\}.
$$

Letting $k \to \infty$, we have:

$$
\limsup_{k \to \infty} M(x_{2n(k)}, x_{2m(k)-1}) = \limsup_{k \to \infty} d(x_{2n(k)}, x_{2m(k)-1}).
$$

From the $b$-triangular inequality, we have:

$$
d(x_{2n(k)}, x_{2m(k)-1}) \leq s(d(x_{2n(k)}, x_{2m(k)-2}) + d(x_{2m(k)-2}, x_{2m(k)-1})).
$$

Letting again $k \to \infty$ in the above inequality, we get:

$$\limsup_{k\to\infty} d(x_{2n(k)}, x_{2m(k)-1}) \leq s\varepsilon. \tag{14}$$

From Condition (13) and Condition (14), we obtain:

$$
\begin{aligned}
\varepsilon &\leq \limsup_{k\to\infty} \left(\beta(M(x_{2n(k)}, x_{2m(k)-1}))M(x_{2n(k)}, x_{2m(k)-1})\right) \\
&= \limsup_{k\to\infty} \beta(M(x_{2n(k)}, x_{2m(k)-1})) \limsup_{k\to\infty} d(x_{2n(k)}, x_{2m(k)-1}) \\
&\leq s\varepsilon \limsup_{k\to\infty} \beta(M(x_{2n(k)}, x_{2m(k)-1})).
\end{aligned}
$$

Therefore,

$$\frac{1}{s} \leq \limsup_{k\to\infty} \beta(M(x_{2n(k)}, x_{2m(k)-1})) \leq \frac{1}{s}.$$

Since $\beta \in \mathcal{B}$, it follows that:

$$\lim_{k\to\infty} M(x_{2n(k)}, x_{2m(k)-1}) = 0.$$

Consequently,

$$\lim_{k\to\infty} d(x_{2n(k)}, x_{2m(k)-1}) = 0. \tag{15}$$

From Condition (11) and using the $b$-triangular inequality, we get:

$$\varepsilon \leq d(x_{2n(k)}, x_{2m(k)}) \leq s(d(x_{2n(k)}, x_{2m(k)-1}) + d(x_{2m(k)-1}, x_{2m(k)})).$$

Letting $k \to \infty$ in the above inequality and using Condition (15), we obtain:

$$\limsup_{k\to\infty} d(x_{2n(k)}, x_{2m(k)}) = 0.$$

This contradicts Condition (11). This implies that $\{x_{2n}\}$ is a $b$-Cauchy sequence and so is $\{x_n\}$. There exists $x^* \in X$ such that $\lim_{n\to\infty} x_n = x^*$. If $T$ is continuous, we have:

$$Tx^* = \lim_{n\to\infty} Tx_{2n} = \lim_{n\to\infty} x_{2n+1} = x^*.$$

From Condition (8), we have:

$$sd(x^*, Sx^*) = sd(Tx^*, Sx^*) \leq \beta(M(x^*, x^*))M(x^*, x^*),$$

where:

$$
\begin{aligned}
M(x^*, x^*) &= max\{d(x^*, x^*), d(x^*, Tx^*), d(x^*, Sx^*)\} \\
&= d(x^*, Sx^*).
\end{aligned}
$$

Since $\beta \in \mathcal{B}$, we have,

$$sd(x^*, Sx^*) \leq \beta(M(x^*, x^*))d(x^*, Sx^*) \leq \frac{1}{s}d(x^*, Sx^*).$$

Hence, $Sx^* = x^*$. If $S$ is continuous, then, by a similar argument to that of above, one can show that $T, S$ have a common fixed point. Now, we prove the uniqueness of the common fixed point. Let $y = Ty = Sy$, is another common fixed point for $T$ and $S$. From Condition (8), we obtain:

$$sd(x^*, y) = sd(Tx^*, Sy) \le \beta(M(x^*, y))M(x^*, y),$$

where:

$$M(x^*, y) = max\{d(x^*, y), d(x^*, Tx^*), d(y, Sy)\} = d(x^*, y).$$

Therefore, $x^* = y$ and the common fixed point $T$ and $f$ is unique.   □

In Theorem 4, if $T = S$, we get the following result.

**Corollary 1.** *Let* $(X, d)$ *be a b-complete b-metric space with parameter $s \ge 1$ and $T$ be self-mapping on $X$ which satisfy:*

$$sd(Tx, Ty) \le \beta(M(x, y))M(x, y), \quad x, y \in X, \tag{16}$$

*where $M(x, y) = max\{d(x, y), d(x, Tx), d(y, Ty)\}$ and $T$ is continuous. Then $T$ has a unique fixed point.*

**Example 3.** *Let $X = [0, 1]$ and $d : X \times X \to [0, \infty)$ be defined by $d(x, y) = |x - y|^2$, for all $x, y \in [0, 1]$. It is easy to check that $(X, d)$ is a b-metric space with parameter $s = 2$. Set $Tx = \frac{x}{4}$ for all $x \in X$ and $\beta(t) = \frac{1}{4}$ for all $t > 0$. Then,*

$$
\begin{aligned}
2d(Tx, Ty) &= 2|\frac{x}{4} - \frac{y}{4}|^2 \\
&\le \frac{1}{4}|x - y|^2 \\
&\le \beta(M(x, y))M(x, y).
\end{aligned}
$$

*Then, the conditions of Corollary 1 are satisfied.*

## 3. Applications to Nonlinear Integral Equations

In this section, we studied the existence of solutions for nonlinear integral equations, as an application to the fixed point theorems proved in the previous section.

Let $X = C[0, l]$ be the set of all real continuous functions on $[0, l]$ and $d : X \times X \to [0, \infty)$ be defined by:

$$d(u, v) = max_{0 \le t \le l}|u(t) - v(t)|^2, \qquad u, v \in X.$$

Obviously, $(X, d)$ is a complete $b$-metric space with parameter $s = 2$. First, consider the integral equation:

$$u(t) = h(t) + \int_0^l G(t, s)k(t, s, u(s)) \, ds, \tag{17}$$

where $l > 0$ and $h : [0, l] \to \mathbb{R}, G : [0, l] \times [0, l] \to \mathbb{R}$ and $k : [0, l] \times [0, l] \times \mathbb{R} \to \mathbb{R}$ are continuous functions.

**Theorem 5.** *Suppose that the following hypotheses hold:*
*(1) for all $t, s \in [0, l]$ and $u, v \in X$, we have:*

$$|k(t, s, u(s)) - k(t, s, v(s))| \le \frac{\sqrt{e^{-M(u,v)}M(u, v)}}{2},$$

*(2) for all $t, s \in [0, l]$, we have:*

$$max \int_0^l G(t,s)^2 \, ds \leq \frac{1}{l}.$$

*Then, the integral equation (see Condition ([17](#))) has a unique solution $u \in X$.*

**Proof of Theorem 5.** Let $T : X \to X$ be a mapping defined by:

$$Tu(t) = h(t) + \int_0^l G(t,s)k(t,s,u(s)) \, ds, \quad u \in X, t, s \in [0, l].$$

From Condition (1) and Condition (2), we can write:

$$
\begin{aligned}
d(Tu, Tv) &= max_{t \in [0,l]} |Tu(t) - Tv(t)|^2 \\
&= max_{t \in [0,l]} \left\{ \left| h(t) + \int_0^l G(t,s)k(t,s,u(s)) \, ds - h(t) - \int_0^l G(t,s)k(t,s,v(s)) \, ds \right|^2 \right\} \\
&= max_{t \in [0,l]} \left\{ \left| \int_0^l G(t,s)(k(t,s,u(s)) - k(t,s,v(s))) \, ds \right|^2 \right\} \\
&\leq max_{t \in [0,l]} \left\{ \int_0^l G(t,s)^2 ds \int_0^l |k(t,s,u(s)) - k(t,s,v(s))|^2 \, ds \right\} \\
&\leq \frac{1}{l} \int_0^l \left| \frac{\sqrt{e^{-M(u,v)} M(u,v)}}{2} \right|^2 ds \\
&\leq \frac{e^{-M(u,v)}}{2} M(u,v).
\end{aligned}
$$

So, we get:

$$d(Tu, Tv) \leq \beta(M(u,v)) M(u,v).$$

Thus, all conditions in Theorem [3](#) for $\beta(t) = \frac{e^{-t}}{2}, t > 0$ and $\beta(0) \in [0, \frac{1}{2})$ are satisfied and hence $T$ has a fixed point. $\square$

Let $X = C[a,b]$ be the set of all real continuous functions on $[a,b]$ and $X$ equipped with the *b*-metric below,

$$d(u,v) = max_{a \leq t \leq b} \{ (|u(t) - v(t)|)^p \}, \quad p > 1, u, v \in X.$$

Then $(X, d)$ is a complete *b*-metric space with parameter $s = 2^{p-1}$. Now, consider the integral equations:

$$u(t) = \int_a^b G(t,s)k_1(t,s,u(s)) \, ds, \tag{18}$$

and

$$u(t) = \int_a^b G(t,s)k_2(t,s,u(s)) \, ds, \tag{19}$$

where $G : [a,b] \times [a,b] \to \mathbb{R}$ and $k_1, k_2 : [a,b] \times [a,b] \times \mathbb{R} \to \mathbb{R}$ are continuous functions.

**Theorem 6.** *Suppose that:*
*(1) For all $t, s \in [a,b]$ and $u, v \in X$, we have:*

$$|k_1(t,s,u(s)) - k_2(t,s,v(s))| \leq \left( \frac{ln(1 + (|u(s) - v(s)|)^p)}{2^{2p-1}} \right)^{\frac{1}{p}}.$$

*(2) For all $t, s \in [a, b]$, we have:*

$$max_{a \leq t \leq b} \int_a^b G(t,s)^q \, ds \leq \frac{1}{(b-a)^{\frac{q}{p}}}, \quad \frac{1}{p} + \frac{1}{q} = 1.$$

*Then the integral equations (Condition (18) and Condition (19)) have a unique common solution.*

**Proof of Theorem 6.** Let $T, S : X \to X$ be mappings defined by:

$$Tu(t) = \int_a^b G(t,s)k_1(t,s,u(s)) \, ds, \tag{20}$$

and

$$Su(t) = \int_a^b G(t,s)k_2(t,s,u(s)) \, ds. \tag{21}$$

From Condition (1) and Condition (2), we have:

$$
\begin{aligned}
d(Tu, Tv) &= max_{a \leq t \leq b}\{(|Tu(t) - Sv(t)|)^p\} \\
&\leq max_{a \leq t \leq b}\{(|\int_a^b G(t,s)k_1(t,s,u(s)) \, ds - \int_a^b G(t,s)k_2(t,s,v(s)) \, ds|)^p\} \\
&\leq max_{a \leq t \leq b}\{(\int_a^b |G(t,s)|(|k_1(t,s,u(s)) - k_2(t,s,v(s))|) \, ds)^p\} \\
&\leq max_{a \leq t \leq b}\{((\int_a^b |G(t,s)^q| \, ds)^{\frac{1}{q}}(\int_a^b (|k_1(t,s,u(s)) - k_2(t,s,v(s))|)^p \, ds)^{\frac{1}{p}})^p\} \\
&\leq max_{a \leq t \leq b}\{(\int_a^b |G(t,s)^q| \, ds)^{\frac{p}{q}}(\int_a^b (|k_1(t,s,u(s)) - k_2(t,s,v(s))|)^p \, ds)\} \\
&\leq max_{a \leq t \leq b}\{(\frac{1}{(b-a)^{\frac{q}{p}}})^{\frac{p}{q}}(\int_a^b (\frac{ln(1 + (|u(s) - v(s)|)^p)}{2^{2p-1}}) \, ds)\} \\
&\leq \frac{1}{(b-a)}(\int_a^b (\frac{ln(1 + d(u,v))}{2^{2p-1}}) ds \\
&\leq \frac{ln(1 + M(u,v))}{2^{2p-1}}.
\end{aligned}
$$

Therefore, we get the following result:

$$
\begin{aligned}
2^{p-1}d(Tu, Tv) &\leq \frac{M(u,v)}{2^p}. \\
&\leq \beta(M(u,v))M(u,v).
\end{aligned}
$$

Hence, all of the hypotheses of Theorem 4 for $s = 2^{p-1}$ and $\beta(t) = \frac{1}{2^p}$ are satisfied. Then $T$ and $S$ have a common fixed point $u \in X$. □

**Author Contributions:** H.F. contributed in conceptualization, methodology, analysis, data curation, original draft writing and editing. D.S. contributed in analysis, data curation. S.R. contributed in conceptualization, methodology, analysis, data curation, writing, review and editing the revision manuscript .

**Funding:** This research received no external funding.

**Conflicts of Interest:** The authors declare no conflict of interest.

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
