# Peer review of "Fixed Point Theorems for Geraghty Contraction Type Mappings in b-Metric Spaces and Applications"

_axioms, doi:10.3390/axioms8010034_

Round 1

Reviewer 1 Report

Shoud be made some corrections.

Author Response

Dear ‎Prof... Thank you for your useful comments and suggestions. We have modified the manuscript accordingly.‎ Best Regards.‎

Reviewer 2 Report

My comments are attached on the review report

Author Response

Dear Prof... Thank you for your useful comments and suggestions. We have modified the manuscript accordingly. Best Regards.

Reviewer 3 Report

I propose the minor revision listed in the attached document. 

Author Response

Dear ‎Prof...‎\\ Thank you for your useful comments and suggestions. We have modified the manuscript accordingly.‎ Best Regards.‎\\

Reviewer 4 Report

The fact that beta is upper bounded by 1/s is enough to obtain the convergence of the sequence, the Cauchy property and even the fact that the limit of the sequence is a fixed point of T. So the property of beta function is not needed in the proof of the results.

Author Response

Dear Prof ...

Thank you for your useful comments and suggestions.

‎According to assuming $s\geq 1$‎.

‎If $s> 1$‎, ‎then we have‎

‎$d(x_n,x_{n+1})<\dfrac{1}{s}d(x_{n-1},x_{n}),$ where $\frac{1}{s}<1$‎. ‎So we concluded that $\{x_n\}$ is a Cauchy sequence‎.

‎But‎, ‎while $s=1$‎, ‎we have $d(x_n,x_{n+1})<d(x_{n-1},x_{n})$‎. ‎Then $\{x_n\}$ is a nonincreasing sequence.

‎Best Regard‎s.

Round 2

Reviewer 4 Report

No comment